# The Role of Metastasectomies and Immunotherapy in the Management of Melanoma Lung Metastases: An Analysis of the National Cancer Database

**DOI:** 10.3390/cancers17020206

**Published:** 2025-01-10

**Authors:** Panagiotis Tasoudis, Vasiliki Manaki, Shannon Parness, Audrey L. Khoury, Chris B. Agala, Benjamin E. Haithcock, Gita N. Mody, Jason M. Long

**Affiliations:** 1Division of Cardiothoracic Surgery, Department of Surgery, School of Medicine, University of North Carolina at Chapel Hill, Chapel Hill, NC 27599, USA; shannon_parness@med.unc.edu (S.P.); audrey.khoury@unchealth.unc.edu (A.L.K.); benjamin_haithcock@med.unc.edu (B.E.H.); gita_mody@med.unc.edu (G.N.M.); jason_long@med.unc.edu (J.M.L.); 2School of Medicine, Aristotle University of Thessaloniki, 54124 Thessaloniki, Greece; vassiamanaki@gmail.com; 3Department of Surgery, School of Medicine, University of North Carolina at Chapel Hill, Chapel Hill, NC 27599, USA; chris_agala@med.unc.edu

**Keywords:** metastasectomy, immunotherapy, melanoma lung metastasis

## Abstract

Metastatic melanoma classically settles in the lung and these patients tend to have poor outcomes. The surgical removal of the lung metastases, termed a metastasectomy, is a well-known treatment option for select patients with this diagnosis. With the introduction of immunotherapy as a treatment option, the aim of this study was to analyze patient survival based on treatment with a metastasectomy or immunotherapy. Using the National Cancer Database, 625 patients were identified as fitting the criteria of this study. We found that during a median follow-up time of 34.6 months, treatment with a metastasectomy significantly improved survival compared to immunotherapy alone. There was no significant difference between treatment with a metastasectomy alone and a combination of a metastasectomy and immunotherapy. Therefore, our findings contribute to the support for surgical treatment in patients with metastatic melanoma to the lungs.

## 1. Introduction

The lung is a frequent location for metastatic tumors to establish themselves since it serves as a sieve for all of the blood that flows through the body [1]. In fact, a total of 18 per 100,000 patients newly diagnosed with cancer present with a synchronous lung metastasis, defined as the diagnosis of a distant lung metastasis together with or within a three-month interval of the diagnosis of the primary tumor [1]. Contemporary data indicate that approximately 100,000 cases of invasive melanoma are reported each year, out of which nearly 5% of cases present with synchronous metastatic disease [2]. Patients with untreated metastatic disease have a 5-year survival of less than 10% [3]. According to the National Comprehensive Cancer Network (NCCN), patients with cutaneous melanoma presenting with oligometastatic disease may receive a surgical resection of the metastasis as a treatment option; however, specific treatment options and guidelines for pulmonary metastases were not mentioned [4].

A pulmonary metastasectomy was first described in 1882 and has been historically advocated in selected patients with metastatic lung disease ever since [5]. Although up to this date, there are no stringent selection criteria on which patients should be considered for a pulmonary metastasectomy [6], a number of reports have shed some light on this treatment modality and have highlighted its benefits [7,8]. A recent review article by Chueng et al. highlights that patients considered for a pulmonary metastasectomy include those that have decent control of their primary disease without wide dissemination, those who can obtain the complete resection of their lung metastasis, those who can tolerate surgery, and those who do not have a better treatment option [9]. It should be noted that in contemporary years, the role of immunotherapy has been heavily investigated in the setting of stage IV melanoma [10,11]. The use of a combination of therapies, which involve inhibitors such as cytotoxic T-lymphocyte-associated antigen 4 and programmed cell death protein 1, has shown excellent response rates so far in patients whose primary disease was completely removed, preventing disease relapse [12,13]. To the best of our knowledge, however, there are no reports assessing the safety and efficacy of immunotherapy as a definitive treatment for a melanoma lung metastasis compared to a metastasectomy.

Taken all together, in such a dynamic landscape as new technologies in the field of immunotherapy are constantly pushing the envelope, it is of paramount significance to have good-quality long-term data to provide to patients and help them in their decision making. Accounting for the fact that the literature revolving around this topic is limited, we decided to conduct this study utilizing the National Cancer Database (NCDB) to identify patients with a melanoma lung metastasis who underwent a metastasectomy with or without immunotherapy, aiming to better define the role of metastasectomies in the armamentarium of thoracic surgeons in the emerging era of immunotherapy-oriented oncology.

## 2. Materials and Methods

### 2.1. Patient Selection

This is a retrospective analysis using the National Cancer Database (NCDB) 2020 version (2004–2020) of deidentified data. This database captures newly diagnosed cancer cases in the United States. The NCDB and its partners are not responsible for any conclusions drawn from the data. This study was approved by the Institutional Review Board at the University of North Carolina (IRB# 20-1493).

The NCDB was queried for adults at least 18 years of age diagnosed with primary melanoma and synchronous lung metastases between the years 2004 and 2015. We excluded all patients who had melanoma metastases to sites other than the lung and patients with multiple distant metastases. This ensured that we did not analyze any cases that did not contribute to our goal to study primary melanoma with a lung metastasis. We used the NCDB variable “Surgical Procedure—Other Site” to define our metastasectomy cohort. Since only patients with a lung-only melanoma metastasis were included in our final synthesis, we believed that it was reasonable to assume that the code that referred to distant tumor resection reflected patients who received a pulmonary metastasectomy.

Ultimately, three groups were generated based on the treatment plan that was followed: (a) the metastasectomy group that included patients who underwent a melanoma metastasectomy but had not received immunotherapy, (b) the immunotherapy group that included patients who had received immunotherapy but did not undergo a surgical resection of the lung metastases, and (c) the metastasectomy and immunotherapy group that included patients whose treatment plan involved both immunotherapy and a resection of the lung metastasis. All the definitions regarding the characteristics of the patients and treatment plans were identified using the Systemic Surgery Sequence and Radiation Surgery Sequence variables.

### 2.2. Statistical Analysis

Categorical variables were expressed as frequencies with corresponding percentages. Continuous variables were expressed as the median [first quartile, third quartile]. The overall survival (OS) was analyzed using the Kaplan–Meier method with the time of diagnosis being timepoint zero. Risk factors, including the age, sex, surgery at the primary site, histology, lymph node status, presence of comorbidities, location of the primary tumor, and treatment facility type, were analyzed in an adjusted and unadjusted multivariate Cox proportional hazards model and summarized using hazard ratios (HRs) and 95% confidence intervals (CIs).

Table 1 and Table 2 evaluate the tumor characteristics and baseline characteristics of the patients identified for our study. A χ^2^ test or Fisher exact test, as appropriate, was used for categorical variables and an analysis of variance was used for continuous variables. In Figure 1, the Cox proportional hazard model was used with the assumption that the risk of death is constant over time. Also, Figure 2 represents the Kaplan–Meier method, which further analyzed the OS while adjusting for age, the Charlson–Deyo score, and where the patient was treated. The statistical software used was STATA 18.0.

## 3. Results

### 3.1. Patients’ Baseline and Tumor Characteristics

A total of 625 patients with a synchronous melanoma lung metastasis who received immunotherapy or underwent a metastasectomy were identified between 2004 and 2015 across the United States (Table 1). From that pool, 280 patients did not receive immunotherapy but had a lung metastasis resection, 267 patients received immunotherapy, and 78 patients received immunotherapy along with a metastasectomy. The majority of the included patients in all three groups were males (69.8%) and were identified as Caucasians (96.6%). In the metastasectomy cohort, 73.9% of the identified patients had a Charlson–Deyo score equal to zero, whereas in the immunotherapy and metastasectomy plus immunotherapy cohorts, the respective percentages were 80.9% and 73.1%. Regarding the tumor characteristics, 4.6% of the tumors in the metastasectomy group, 5.6% of the tumors in the immunotherapy group, and 3.6% of the tumors in the metastasectomy plus immunotherapy group were found to be spindle cell melanomas (Table 2). A total of 53.2% of the included patients in the metastasectomy cohort had positive lymph nodes, whereas 71.9% and 70.5% of the included patients had positive lymph nodes in the immunotherapy and metastasectomy plus immunotherapy cohorts, respectively (Table 2).

### 3.2. Treatment

In terms of treatment, surgery at the primary site was performed in 23.6% of the patients in the metastasectomy cohort, 41.2% of the patients in the immunotherapy cohort, and 33.3% in the metastasectomy plus immunotherapy cohort, respectively (Table 2). Moreover, only 4.6%, 12.0%, and 9.0% of the patients in the metastasectomy, immunotherapy, and metastasectomy plus immunotherapy cohorts, respectively, received radiation therapy. Finally, the majority of patients were treated in academic hospitals (49.0%) and in metropolitan areas (79.4%).

### 3.3. Overall Survival

The median follow-up time in the metastasectomy cohort was 34.9 [interquartile range (IQR): 14.5, 84.2] months and the 1- and 5-year OS was 80.4% and 45.2%, respectively. The median follow-up time was 28.1 [10.5, 69.8] and 55.3 [18.1, 86.2] months in the immunotherapy and metastasectomy plus immunotherapy groups, while the 1- and 5-year OS was 77.4% and 37.0% in the immunotherapy cohort and 88.3% and 56.9% in the metastasectomy plus immunotherapy cohort, respectively (Figure 1).

When we used unadjusted models, a metastasectomy plus immunotherapy was found to offer better survival outcomes compared to metastasectomy alone (unadjusted hazard ratio [HR]: 0.63; 95% confidence interval [CI]: 0.44–0.91; *p* = 0.01), whereas immunotherapy alone was not found to offer significantly different outcomes to a metastasectomy (HR: 1.18; 95% CI: 0.96–1.45; *p* = 0.13). When we adjusted our analyses for all the aforementioned risk factors, we found that immunotherapy alone was linked to worse OS compared to a metastasectomy (HR: 1.32; 95% CI: 1.04–1.67; *p* = 0.02) and that the addition of immunotherapy to a metastasectomy does not offer better OS compared to a metastasectomy alone (HR: 0.75; 95% CI: 0.51–1.10; *p* = 0.14). Other factors that were found to be associated with worse survival outcomes according to our adjusted survival analytic models were an increased age, high Charlson–Deyo comorbidity score, and treatment in a community cancer center (Table 3). The OS curves of the three comparative groups adjusted for all the aforementioned variables are depicted in Figure 2.

## 4. Discussion

The outcomes of this study suggest that a lung metastasectomy may be beneficial in terms of the overall survival (OS) for patients with melanoma lung metastases. When the analysis was adjusted for factors such as the age, sex, primary surgery, histology, lymph node status, comorbidities, primary tumor location, and type of treatment facility, immunotherapy alone was not associated with improved OS in this patient population compared with patients who were treated with a metastasectomy. Additionally, in an unadjusted model analysis, the combination of a metastasectomy and immunotherapy appeared superior to a metastasectomy alone; however, when we adjusted our analyses for all of the aforementioned risk factors, the combination of the treatment strategies did not appear to significantly improve the outcomes. This may be due to our adjusted risk factors proving to be confounding variables within our study as significance was lost once adjusted.

Even though numerous articles have been published since the first pulmonary metastasectomy, there is still a scarcity of data on this topic [14,15,16]. The most significant report to date is from the International Registry of Lung Metastases, which includes 5,206 patients with lung metastases [17]. According to this study, patients who underwent metastasectomies had survival rates of 36%, 26%, and 22% at 5, 10, and 15 years, respectively [17]. In their study of 1720 patients with pulmonary metastatic melanoma, Petersen et al. found out that a pulmonary metastasectomy was beneficial for a subset of patients with several characteristics, such as those with fewer than two pulmonary metastases, no disease recurrence, or an extrathoracic pulmonary metastasis [8]. Other previous studies in the literature have also demonstrated the beneficial effects of a metastasectomy for selected patients [7,18]. However, due to the limited amount of concrete evidence, guidelines on the topic are primarily based on the opinions of expert groups who have discussed and provided recommendations on when a pulmonary metastasectomy should and should not be considered.

Previous reports and recommendations from surgical societies have suggested that a a metastasectomy should be performed in patients with isolated pulmonary metastases in whom the primary disease is controlled or controllable and there are no other systemic metastases, or if present, they are being actively managed [6,19,20]. This statement, however, is mostly based on experts’ opinions rather than solid evidence and is generalized to all pulmonary metastases irrespective of the primary tumor site and histology. Interestingly, our results showed that surgery at the primary site does not significantly influence long-term survival, but this finding might be underpowered by the low number of identified patients in all three comparative groups.

It should be noted that data regarding immunotherapy from the previous decade were not encouraging for melanoma treatment. The documented rates of positive responses have consistently fallen below 20% for various types of immunotherapy, such as cytokines, monoclonal antibodies, and vaccination methods using synthetic peptides, naked DNA, dendritic cells, recombinant viruses, and similar techniques [21]. In this context, chemotherapy with dacarbazine used to be the mainstay treatment for advanced melanoma, and it is now the only licensed chemotherapeutic drug in most countries for treating metastatic melanoma [22,23]. Of note, other medications, including temozolomide, paclitaxel, docetaxel, cis/carboplatin, and nitrosoureas, have been used off-label with questionable outcomes [23]. The emergence of immunotherapy in our era has led to a paradigm shift in the management of advanced melanoma. The IMMUNED trial, a randomized, double-blind, placebo-controlled phase 2 trial conducted by 20 German academic healthcare institutions, provided strong evidence that patients with stage IV disease, regardless of their BRAF V600 mutation status, who have undergone the primary definitive treatment of all sites of disease (either surgery or radiation therapy) benefit significantly from adjuvant immunotherapy [12,13].

Our study aimed to evaluate the effect of immunotherapy alone versus when combined with a lung metastasectomy in patients in whom the primary and/or the metastatic disease are not entirely removed on the overall survival. In this setting, it seems that the addition of immunotherapy does not significantly improve survival compared to a metastasectomy, and our results emphasize the importance of surgical tumor resection. This could be explained by the reasonable speculation that the patients in the immunotherapy cohort could be critically ill patients, patients with a higher disease burden who were not eligible for the surgical resection of their metastasis, patients with a poor response to immunotherapy, and those who were not deemed eligible for the resection of their metastasis even after immunotherapy was administered. These factors could not be explicitly evaluated using the NCDB, but our data showed a significant difference in the amount of positive lymph node involvement, alluding to more extensive disease, between the immunotherapy-alone group and our surgical groups. Importantly, our results showed that the addition of a lung metastasectomy to treatment regimens in this patient population improved the overall survival. While the addition of immunotherapy to the treatment regimen of metastatic melanoma is a crucial one, our study found that a metastasectomy alone and a metastasectomy combined with immunotherapy had improved survival rates compared to immunotherapy alone and therefore should be recognized as an important treatment option for patients with lung metastases. The NCCN guidelines mention surgical resection as a treatment option for oligometastatic disease as a whole but do not mention specifics for the treatment of pulmonary metastases [4]. Future randomized control studies should be performed to test the hypothesis that there is no survival benefit to receiving both a metastasectomy and immunotherapy compared to a metastasectomy alone.

The IMMUNED trial assessed the efficacy of adjuvant therapy with nivolumab, ipilimumab, or a combination in patients with melanoma and demonstrated the efficacy of immunotherapy in terms of disease relapse prevention but found no difference in the overall survival between nivolumab and a placebo [13]. The final results of this study were published in 2022 and showed that adjuvant therapy with ipilimumab and nivolumab significantly increased the overall survival compared to a placebo [12]. In our study, the addition of immunotherapy to treatment regimens in patients who underwent the resection of their metastatic disease did not offer a benefit regarding the long-term OS compared to patients who were only treated with a metastasectomy. Unfortunately, we could not assess the impact of immunotherapy on the recurrence rates and the disease-free survival since the NCDB does not capture the disease recurrences.

Finally, there are several other limitations in this study that should be acknowledged. To begin with, specific data were not available in the NCDB and could not be assessed. There was no mention of the number of lung metastases, the mediastinal lymph node involvement, the type of immunotherapy drugs administered, and the timing of immunotherapy and surgery, nor of serum lactose dehydrogenase (LDH) levels. Serum LDH is another prognostic factor for patients with metastatic melanoma, and the American Joint Committee on Cancer (AJCC) states that increasing levels are linked to poorer prognoses [24]. The time between the diagnosis and the metastasectomy, the extent of the metastatic tumor’s excision, and any postoperative issues were also unclear. These limitations restricted our ability to fully evaluate how the disease burden impacted patient outcomes. Lastly, since there was no access to individual patient data, the impact of specific variables on the outcomes could not be investigated. In particular, the absence of detailed information on the type of surgery performed, the status of surgical margins, and post-surgical complications prevented us from analyzing the impact of these variables on patient outcomes. We recognize the inherent biases involved without being able to evaluate these variables. However, using the data that were available to us, we used a risk-stratified statistical model to control as much bias as possible. Another area of limitation is the timeframe of our data collection. Immunotherapies have changed greatly since 2010 with the creation of better and more successful treatments, so it is possible our study included patients who were treated with therapies that are no longer used in the standard regimen and had lower response rates than current options. For example, per the United States Food and Drug Administration (FDA) website, for unresectable or metastatic melanoma, single-use ipilimumab was first approved in 2011, single-use nivolumab and single-use pembrolizumab in 2014, ipilimumab plus nivolumab in 2015, and finally nivolumab plus relatlimab in 2022 [25]. Therefore, our cohort included patients who were likely treated with regimens that are now considered outdated and less effective.

## 5. Conclusions

Our study examined patients with melanoma lung-only metastases from 2010 to 2015 across the United States. Our findings indicate that a metastasectomy is associated with better overall survival compared to immunotherapy alone. Although there is strong evidence supporting the efficacy of immunotherapy in preventing disease relapse in patients with completely resected/eradicated disease, our findings suggest that the addition of immunotherapy to treatment regimens in patients who underwent resection of the lung metastasis does not seem to significantly improve the overall survival. The inherent limitations of a retrospective study include the difficulty to assign causality due to the lack of randomization and the existence of possible confounders. Additionally, without being able to analyze the type of immunotherapy used for each patient, conclusions are difficult to draw between our results and the current immunotherapy regimen. However, our data may assist with a comparison between previous and current treatment options for metastatic melanoma while assessing the benefits of a pulmonary metastasectomy. Within our dataset, there was a correlation between the treatment method and overall survival. Therefore, our results emphasize the importance of surgical tumor resection at the metastatic site, and we exercise caution in expanding this conclusion to current immunotherapy options with a pulmonary metastasectomy.

## Figures and Tables

**Figure 1 cancers-17-00206-f001:**
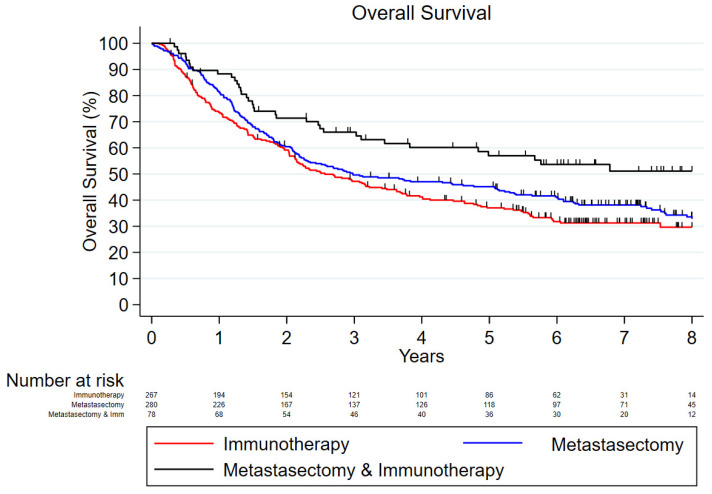
Unadjusted Cox proportional hazards model measuring overall survival within each treatment group.

**Figure 2 cancers-17-00206-f002:**
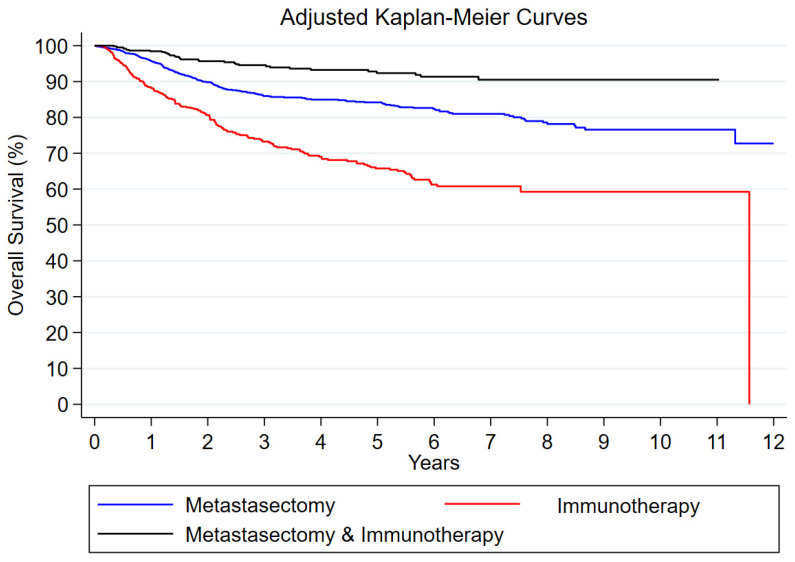
Adjusted Kaplan–Meier curves for age, Charlson–Deyo score, and treatment center location for overall survival within each treatment group.

**Table 1 cancers-17-00206-t001:** Patients’ baseline characteristics compared within each treatment group.

	Metastasectomy	Immunotherapy	Metastasectomy and Immunotherapy	*p*-Value
Total Number	280	267	78	
Age	69 [59.50, 77]	65 [55, 76]	64.5 [52, 71]	0.34
Gender				0.42
Male	201 (71.79)	185 (69.29)	50 (64.10)	
Female	79 (28.21)	82 (30.71)	28 (35.90)	
Ethnic Group				0.86
Caucasian	270 (96.43)	256 (95.88)	78 (100)	
African American	2 (0.71)	3 (1.12)	0 (0)	
Other	8 (2.87)	8 (2.97)	0 (0)	
Year of Diagnosis				<0.0001
2010	54 (19.29)	14 (5.24)	6 (7.69)	
2011	57 (20.36)	20 (7.49)	13 (16.67)	
2012	50 (17.86)	25 (9.36)	13 (16.67)	
2013	43 (15.36)	55 (20.60)	12 (15.38)	
2014	41 (14.64)	63 (23.60)	13 (16.67)	
2015	35 (12.50)	90 (33.71)	21 (26.92)	
Type of Insurance				0.03
Not Insured	6 (2.14)	9 (3.37)	1 (1.28)	
Private Insurance/Managed Care	103 (36.79)	100 (37.45)	39 (50)	
Medicaid	11 (3.93)	18 (6.74)	1 (1.28)	
Medicare	159 (56.79)	131 (49.06)	36 (46.15)	
Other Government	1 (0.36)	4 (1.50)	1 (1.28)	
Insurance Status Unknown	0 (0)	5 (1.87)	0 (0)	
Median Income Quartiles 2012–2016				0.69
<$40,227	30 (10.70)	24 (9)	6 (7.70)	
$40,227–$50,353	51 (18.20)	42 (15.70)	12 (15.40)	
$50,354–$63,332	70 (25)	57 (21.30)	14 (17.90)	
>=$63,333	94 (33.60)	97 (36.30)	33 (42.30)	
N/A	35 (12.50)	47 (17.60)	13 (16.70)	
Charlson–Deyo Score				0.42
0	207 (73.93)	216 (80.90)	57 (73.08)	
1	58 (20.71)	36 (13.48)	15 (19.23)	
2	10 (3.57)	9 (3.37)	4 (5.13)	
3	5 (1.79)	6 (2.25)	2 (2.56)	

Abbreviations: N/A: not available. All values are reported as frequencies (corresponding %) or medians [first quartile, third quartile].

**Table 2 cancers-17-00206-t002:** Tumor characteristics and treatment details compared within each treatment group.

	Metastasectomy	Immunotherapy	Metastasectomy and Immunotherapy	*p*-Value
Total Number	280	267	78	
Histology				0.08
Melanoma, NOS	233 (83.21)	193 (72.28)	61 (78.21)	
Nodular Melanoma	13 (4.64)	27 (10.11)	8 (10.26)	
Superficial Spreading	8 (2.86)	17 (6.37)	2 (2.56)	
Desmoplastic Melanoma	2 (0.71)	7 (2.62)	2 (2.56)	
Spindle Cell Melanoma	13 (4.64)	15 (5.62)	3 (3.85)	
Other	11 (3.93)	8 (3)	2 (2.56)	
Lymph Nodes				<0.001
Negative Lymph Nodes	39 (13.93)	7 (2.62)	3 (3.85)	
Positive Lymph Nodes	149 (53.22)	192 (71.91)	55 (70.51)	
Not Assessed	92 (32.86)	68 (25.47)	20 (25.64)	
Location				<0.001
Head and Neck	25 (8.92)	46 (17.22)	8 (10.26)	
Trunk	18 (6.43)	42 (15.73)	9 (11.54)	
Upper Limb	19 (6.79)	31 (11.61)	8 (10.26)	
Lower Limb	6 (2.14)	24 (8.99)	5 (6.41)	
Skin, NOS	212 (75.71)	124 (46.44)	48 (61.54)	
Surgery at Primary Site				<0.001
No	214 (76.43)	157 (58.8)	52 (66.67)	
Yes	66 (23.57)	110 (41.20)	26 (33.33)	
Radiation				<0.001
No	267 (95.36)	235 (88.01)	71 (91.03)	
Yes	13 (4.64)	32 (11.99)	7 (8.97)	
Facility Type				0.10
Community Cancer Program	9 (3.20)	14 (5.20)	0 (0)	
Comprehensive Community Cancer Program	84 (30)	68 (25.50)	15 (19.20)	
Academic/Research Program	131 (46.80)	132 (49.40)	43 (55.10)	
Integrated Network Cancer Program	50 (17.90)	39 (14.60)	17 (21.80)	
N/A	6 (2.10)	14 (5.20)	3 (3.80)	
Facility Location				0.09
Metro	222 (79.29)	212 (79.4)	62 (79.49)	
Urban	45 (16.07)	35 (13.11)	9 (11.54)	
Rural	3 (1.07)	9 (3.37)	0 (0)	
N/A	10 (3.57)	11 (4.12)	7 (8.97)	
Interval from Dx to Immunotherapy	N/A	64 [39, 97]	84 [57, 123]	0.06
Follow-up	34.9 [14.50, 84.20]	28.1 [10.50, 69.80]	55.3 [18.10, 86.20]	0.001

Abbreviations: NOS: Not otherwise specified; N/A: not available; Dx: diagnosis. All values are reported as frequencies (corresponding %) or medians [interquartile ranges].

**Table 3 cancers-17-00206-t003:** Unadjusted and adjusted estimated hazard ratios with confidence intervals comparing tumor characteristics and treatment details within each treatment group.

	Unadjusted Estimates	95% CI			Adjusted Estimates	95% CI		
Characteristic	HR	LCL	UCL	*p*-Value	HR	LCL	UCL	*p*-Value
Gender								
Male (reference)								
Female	0.83	0.66	1.03	0.10	0.87	0.69	1.10	0.24
Age	1.02	1.01	1.03	<0.001	1.02	1.01	1.03	<0.001
Treatment Plan								
Metastasectomy (reference)								
Immunotherapy	1.18	0.96	1.45	0.13	1.32	1.04	1.67	0.02
Metastasectomy and Immunotherapy	0.63	0.44	0.91	0.01	0.75	0.51	1.10	0.14
Surgery at Primary Site	0.84	0.68	1.04	0.11	0.81	0.55	1.20	0.30
Histology								
Melanoma, NOS (reference)								
Nodular Melanoma	0.83	0.56	1.23	0.36	1.07	0.67	1.69	0.78
Superficial Spreading	1.06	0.66	1.68	0.82	1.54	0.90	2.66	0.12
Desmoplastic Melanoma	1.11	0.52	2.34	0.79	1.25	0.56	2.78	0.58
Spindle Cell Melanoma	0.69	0.42	1.14	0.15	0.79	0.47	1.32	0.37
Other	1.12	0.66	1.88	0.68	1.43	0.82	2.49	0.21
Lymph nodes								
Negative (reference)								
Positive	1.13	0.86	1.49	0.39	1.07	0.77	1.49	0.70
Not Assessed	1.24	0.99	1.55	0.06	1.24	0.98	1.58	0.07
Charlson–Deyo Score								
0 (reference)								
1	1.16	0.90	1.50	0.25	1.21	0.92	1.58	0.17
2	1.16	0.70	1.92	0.56	1.07	0.64	1.80	0.80
3	2.84	1.62	4.96	<0.001	2.54	1.43	4.50	0.001
Location								
Skin, NOS (reference)								
Head and Neck	0.95	0.70	1.29	0.76	0.99	0.64	1.54	0.96
Trunk	0.89	0.64	1.23	0.48	0.95	0.57	1.56	0.83
Upper Limb	0.75	0.51	1.09	0.13	0.84	0.51	1.39	0.50
Lower Limb	1.12	0.74	1.68	0.60	1.17	0.68	2.01	0.56
Facility Type								
Academic/Research Program (reference)								
Community Cancer Program	1.54	0.91	2.62	0.11	1.22	0.70	2.11	0.49
Comprehensive Community Cancer Program	1.57	1.25	1.99	<0.001	1.46	1.15	1.84	0.002
Integrated Network Cancer Program	1.17	0.88	1.56	0.27	1.15	0.86	1.54	0.34

Abbreviations: HR: hazard ratio; 95% CI: 95% confidence interval; LCL: lower confidence level; UCL: upper confidence level.

## Data Availability

Restrictions apply to the availability of these data. Data were obtained from the National Cancer Database (NCDB) and are available from the authors with the permission of the NCDB. The National Cancer Database (NCDB) is a joint project of the Commission on Cancer (CoC) of the American College of Surgeons and the American Cancer Society. The CoC’s NCDB and the hospitals participating in the CoC’s NCDB are the source of the deidentified data used herein; they have not verified and are not responsible for the statistical validity of the data analysis or the conclusions derived by the authors.

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
