# Peer review of "The Role of Metastasectomies and Immunotherapy in the Management of Melanoma Lung Metastases: An Analysis of the National Cancer Database"

_cancers, 2025, doi:10.3390/cancers17020206_

Round 1
Reviewer 1 Report (New Reviewer)
Comments and Suggestions for Authors
The authors have analyzed the outcomes of patients coded as having melanoma lung metastases treated with surgical resection only; immunotherapy only; or both. Surprisingly, the analysis indicates best overall survival in patients treated with both modalities, and the worst outcomes in patients treated with immunotherapy alone. This is quite surprising as previous studies of anti-PD1 immunotherapy had identified excellent outcomes in patients with lung metastases only (i.e. PMID 29685882; 1 year survival rate of 89%). The results of this study will likely spur additional investigation.
1. Per the methods section, the authors included patients diagnosed with lung metastases from 2004 - 2015. However, per Table 1, the analysis included patients diagnosed from 2010 - 2015. Why is there a discrepancy between the dates?
2. The date range of 2010 - 2015 is significant due to the immunotherapies that were available during this time. Ipilimumab was the first immunotherapy to improve survival in patients with stage IV melanoma; it was approved in March, 2011. Despite its approval, it was known to have a low response rate (~10% in early studies, up to 20% in later studies), and it rapidly faded from use as first line treatment after the approval of the anti-PD1 antibodies nivolumab (July 2014) and pembrolizumab (September 2014), and further after the approval of ipilimumab + nivolumab (November 2015)- each of which was shown to improve OS versus Ipilimumab alone in stage IV patients. Based on the dates of diagnosis in Table 1, it is likely that this cohort likely primarily included metastatic melanoma patients treated with Ipilimumab, which is no longer a standard 1st line therapy for patients with metastatic melanoma. This is a key limitation of the study in terms of its relevance to the management and outcomes of contemporary metastatic melanoma patients. The context of the dates above, and this limitation, need to be included as important limitations of this study in the Discussion, particularly if the actual treatments received are not available in the database. I would suggest that the outcomes of patients diagnosed before 2011 are either not relevant for this study- or could serve as a before-immunotherapy control.
3. The authors note in the Discussion that adjuvant treatment of stage IV patients with nivolumab did not improve overall survival in the IMMUNED trial.
(a) It should be acknowledged that in that same trial that adjuvant treatment with Ipilimumab + Nivolumab did significantly improve the survival of patients- and that the cohort in this manuscript likely included very few if any patients treated with that regimen. Also would note that the adjuvant PD1 trials in which placebo (i.e. surgery alone) was used in the control arm in patients with stage II and stage III melanoma have demonstrated significant improvements in relapse free survival but no OS results have yet to be reported.
(b) Recent clinical trials (PMID 36856617 and PMID 38828984) have demonstrated in stage III melanoma patients that neoadjuvant immunotherapy is much more effective that adjuvant therapy, consistent with preclinical studies. Thus, it would be interesting to know for the patients treated with surgery and immunotherapy if there was a difference in outcomes based on the sequence of the 2 modalities- although in clinical practice we rarely recommend surgery in patients that had a positive response to immunotherapy treatment.
Author Response
Thank you very much for taking the time to review this manuscript. Please find the detailed responses below and the corresponding in track changes in the re-submitted files.
The discrepancy arises due to the database version used for our analysis. The version available for melanoma data covered the years 2004–2015. However, we were only able to identify patients with lung metastases specifically from 2010–2015 due to limitations in the data coding or availability during that period. This reflects the timeframe within which the data was most accurately and reliably captured for our study. Lines 656-660 were added for the limitation of the timeframe of our study. Unfortunately, without knowing in our data set if the immunotherapy was adjuvant or neoadjuvant, it is difficult to highlight that in our discussion. However, in lines 646-647, we added the limitation that timing of treatment is not known. We added in the acknowledgement of the IMMUNED trial finding improvement in survival of patients in lines 629-631.
Reviewer 2 Report (New Reviewer)
Comments and Suggestions for Authors
The authors examined the role of lung metastasectomy on survival in patients with advanced melanoma. Despite the inclusion of patients from a national registry and the long period of follow-up, there are some issues needed to be addressed by the authors.
Introduction: OK
Methods: -First, patient population should be described in a more clear and analytic way. It is not clear if de novo metastatic melanoma with lung metastasis only were included, or if primary melanomas that metastasize to the lungs were included in the analysis. Also, the extent of disease (such as solitary or multiple lung mets) should be reported. In addition, as the primary endpoint of the study was OS, did patients in the metastasectomy arm receive IO during the course of their disease, or were there any available information regarding the timing of use of immunotherapy and surgery (sequential? Concurrent? and which strategy used first? In concordance, exclusion criteria should be described in a more analytic way.
-Another important point needed to be addressed and analyzed, especially in discussion, is the type of ICI used. Looking at OS curve, the 3y or 5y OS of patients treated with immunotherapy is lower compared to reported in CheckMate067 for nivo/ipi, pinpointing the importance of knowing the type of ICI used. Were there available information regarding the type of ICI?
Discussion: - In my view, the most important finding of the study is the addition of metastasectomy to immunotherapy that lead to a survival benefit. This is the finding that should be discussed in a more thorough way, rather than the comparison of metastasectomy to immunotherapy, especially if information such as type of ICI or extent of lung disease are missing. In concordance, factors included in the adjusted models, such as primary surgery or lymph node status are not deemed prognostic in distant metastatic melanomas.
- s213-220 need revision. Was the aim of the study to evaluate the efficacy of immunotherapy or the role of lung metastasectomy on survival? Also, the authors should state if there were available information regarding factors reported in s217-220. Otherwise, missing information on those factors, could cause imbalance on baseline characteristics, and another approach (i.e. matched case-control, or propensity score matching) should be used.
In general, the results of the study are very interesting and important showing the beneficial role of combining metastasectomy and immunotherapy, however, there are some issues needed to be addressed.
Author Response
Thank you very much for taking the time to review this manuscript. Please find the detailed responses below and the corresponding in track changes in the re-submitted files.
We changed the line describing patient selection to say it was primary melanoma with lung metastases to make it more clear. Unfortunately, the NCDB database did not report on extent of disease (solitary or multiple lung mets) nor the type of immunotherapy used/sequence of treatment, so we cannot report on this. However, we did mention this limitation in lines 644-647 and I added a line about sequence of treatment not being available as well. I included lines 92-93 to make our exclusion criteria clearer as well.
Lines 603-605 were edited to show our aims of the study. Also, lines 612-625 were added to further discuss the possible factors describing our findings that surgical resection had improved OS and increase the discussion of our main findings that the addition of metastectomy improved overall survival.
Reviewer 3 Report (New Reviewer)
Comments and Suggestions for Authors
The authors presented original research about the role of pulmonary metastasectomy versus immunotherapy or surgery plus immunotherapy in metastatic malignant melanoma.
This article has the potential to be relevant to the readers.
These are my points to reconsider:
1. NCCN guidelines offer up-dated scientific information about oligometastatic disease, last version 3.2024, so the reference 3 is completely outdated and must be changed.
2. There is no relevant medical information presented, about the primary tumor, such as TNM stage, risks factors (LDH, margins of resection of primary tumor, Breslow, Clark, BRAF status, adjuvant or palliative treatment until M1PUL).
3. There is no data about which kind of immunotherapy or targeted therapy (for BRAF mutated tumors) was used during the study.
4. Since the time of pulmonary metastasectomy described in references 6,7,13 a lot of things changed in metastatic malignant melanoma and the actual discussion becomes out-of-fashion.
5. The discussion should take into consideration the actual NCCN guidelines to properly integrate these results into practical clinical judgment.
6. Nevertheless, the actual stream of treating malignant melanoma is changing, so being accurate about references and an in-depth discussion are mandatory for reading audience, other wise these results might become misleading.
Author Response
Thank you very much for taking the time to review this manuscript. Please find the detailed responses below and the corresponding in track changes in the re-submitted files.
Unfortunately, the NCDB does not provide characteristics of the primary tumor nor type of immunotherapy. We added this as a limitation in lines 644-647. We included a line about the NCCN guidelines within our discussion (lines 621-622) and within our introduction (lines 52-56). Also, we added a more recent reference evaluating the use of pulmonary metastectomy, included in lines 61-65.
Reviewer 4 Report (New Reviewer)
Comments and Suggestions for Authors
This review is primarily a statistical one, with an overarching recommendation and specific major and minor points. My comments to the authors are as follows:
1. I enjoyed reading this manuscript and learned a lot about the findings related to the predictors of survival in patients with melanoma lung metastases only and their relevance to health research.
2. The introduction section justified why the study was necessary and linked the aims of the study to the readers.
3. The statistical intensity of the manuscript was above the average for articles published in general medical journals. The quality of statistical reporting was week: I scored 3 on a scale of 0 (poor) to 10 (very high). You should improve the quality of statistical reporting and data presentation. See comments below.
4. The Methods section included a subsection entitled Statistical analysis. According to general guidelines for statistical reporting, the main goal of this subsection is to help a knowledgeable reader to judge the appropriateness of the methods for the study and to verify the reported methods. This part of your manuscript is incomplete, and you should improve the description of the statistical methods.
5. Help your readers by providing more information about the data analysis. In the Statistical Analysis section, you need to explain which statistical methods or tests were used for each analysis. In a well-written statistical analysis subsection, authors should also identify the variables used in each analysis. Readers should be informed which variables were analyzed with each method. Care should be taken to ensure that all methods are listed. The statistical section should be consistent with the results section. Examples of items to be revised:
5.1. Describe the objectives and significance test used in Tables 1 and 2.
5.2. Identify the explanatory variables used in the multivariable Cox regression analyses.
5.3. Describe whether the data satisfy the assumptions and prerequisites of the Cox regression method.
5.4. Consider using the log-rank test (or a more appropriate test if you want to emphasize the difference at the beginning of the follow-up period) to assess the statistical significance between the survival curves.
5.5. Good scientific writing includes references and brief descriptions for methods that have been published but are not well known or commonly used. Consider providing statistical references for the method used in Figure 2.
5.6. Include the statistical software used in the analysis. Readers need to know how the results were obtained.
6. Tables 1 and 2 give readers an idea of the extent to which the study results can be generalized to their own local situation. However, the significance test that provides the p-values has not been reported in the tables.
7. Tables and figures should be able to stand on their own. Well-prepared tables and figures with appropriate titles, clear labels, and optimally presented data will enable readers to scrutinize the data. That is, all information necessary for interpretation should be included in titles, tables or figures, legends, or footnotes. This means naming descriptive statistics, significance tests, and multivariable modeling methods used. Below is a list of presentation problems that may prevent readers from quickly grasping the message and reduce the overall quality of the data presentation:
7.1. Identify the significance test used in Tables 1 and 2.
7.2. In order to assess the validity of the results, the reader needs to know the number of study participants. Sample size is an important aspect of research. The total number of participants or the sample size of each group should be clearly stated in all tables and figures. Please include this valuable information in Table 2, Table 3, and Figures 1 and 2. It shows that you have focused on the presentation of your data.
7.3. Name the multivariable modeling method used in Table 3.
7.4. Tables and figures did not have clear titles.
7.5. The repetition of the term "reference" in each column is a bit distracting when there are multiple columns. Please consider replacing it with the number "1" since the regression coefficient in the reference group is equal to one.
7.6. Figure 1: Please provide a log-rank test to assess the statistical significance between the survival curves.
8. Discussion: You have helped your readers well by comparing your results with previous studies on the same topic. In addition, I think readers will appreciate that you did not make unacceptable claims that your study did not support.
Author Response
Thank you very much for taking the time to review this manuscript. Please find the detailed responses below and the corresponding in track changes in the re-submitted files.
Lines 120-130 were added under the statistical analysis section to further describe our methods to address points 4, 5, 6, and 7. Also, we added more detailed titles to each figure and table. The total number of participants was added to table 2 and is represented by timepoint year 0 in figure 1. We used Cox proportional hazard ratios for our survival analysis. The “reference” was removed from table 3 and placed next to their respective items within the first column.
Reviewer 5 Report (New Reviewer)
Comments and Suggestions for Authors
There is no doubt that this paper is of great clinical interest. Nevertheless, I have some comments and suggestions for improvement.
2.2 Statistical analysis: In table 1, 3 groups are compared. However, the statistical tests which have been used are not specified. Some parameters in table 1 (i.e. year of diagnosis, Income, Charlson-Deyo Score) are ordinally scaled. I am not sure which test has been used for these parameters. In any case, Chi2-test is not appropriate (Kruskal test would be better).
The Ethnic group "white" is called "Caucasians" in the tesxt (Section 3.1). This is not consistent.
The number of decimal places should be identital (for all 3 groups, in the tables and in the text). Some percentages are given with two decimal places, others with three.
3.3 Overall survival: The risk factors which have been considered for the adjusted models should be specified. Authors should interpret the results more cautiously. A p value more than 0.05 does NOT mean that the outcomes of two groups are "equivalent". When comparing immunotherapy alone with metastasectomy alone, immunotherapy is linked to worse Overall survival (HR = 1.32, p = 0.024, adjusted model). However, when comparing "metastasectomy+immunotherapy" versus "immunotherapy alone" the HR = 0.75 suggests that "metastasectomy+immunotherapy" is superior. The fact that this result is not significant maybe due the small sample size of only n=78 for the combined therapy. The inverse of 0.75 is 1.33 (very close to HR = 1.32 for the immunotherapy).
Furthermore: When using the unadjusted model, "metastasectomy alone"and "metastasectomy+immunotherapy" differ significantly. This may be caused by a confounder. What is the reason that there is no significance when using the adjusted model?
I think the lines "reference" are not necessary. It is sufficient just to mention the reference group.
Discussion: I think it would be a good idea to encourage further studies (preferably an RCT) to obtain sufficient data to test the hypothesis "There is no benefit of"metastasectomy + immunotherapy" compared to "metastasectomy alone".
Author Response
Thank you very much for taking the time to review this manuscript. Please find the detailed responses below and the corresponding in track changes in the re-submitted files.
We changed the term “white” to “Caucasian” in table 1. We also made every decimal within the tables to 2 places, excepting whole numbers and when the two decimals contained zero prior to another digit. Percentages within the body of the paper were kept to one decimal place. Also, the references were removed from table 3 and placed next to their respective items within the first column.
The risk factors were specified and added in lines 120-121. Also, lines 124-130 were added to further describe our statistical analysis. Also, we changed line 186 to reflect non-significance of the findings. The possibility that the adjusted risk factors were due to a confounder within our risk factors was added in lines 557-558. Finally, we added a lines 623-625 to encourage future randomized studies to test this hypothesis.
Round 2
Reviewer 1 Report (New Reviewer)
Comments and Suggestions for Authors
Thank you for revising the manuscript, but I think that the conclusions are still misleading and that the manuscript needs further editing to be appropriate for publication.
As the authors state, "The objective of this study was to determine predictors of survival for patients with melanoma metastatic to the lung in the era of immunotherapy." Further, the authors state, "it is of paramount significance to have good quality long-term data to provide to the patients and help them in their decision making."
1. The key problem with the manuscript remains that conclusions are being drawn without knowing what immunotherapies the patients received- and particularly the likelihood that the immunotherapies received by the patients in this cohort are irrelevant to contemporary patients. As a result, the conclusions made by the authors may be inaccurate in determining what is the relative efficacy of melanoma lung metastectomy "in the era of immunotherapy." Importantly, making conclusions on data that isn't "good" may lead current practicioners to make decisions about their patients that aren't truly evidence based and potentially harmful to patients.
As I explained in my initial review, based on the dates of diagnosis of the cohort, I believe that most patients treated with immunotherapy likely received single-agent ipilimumab, which has a response rate in stage IV patients of 10-15%- which is basically in the same range as the therapies that the authors noted to be part of the previous era of melanoma treatments in the paragraph that starts on line 217. While the authors do have a single sentence at the end of the Discussion acknowledging this to a degree ("Immunotherapies have changed greatly since 2010 with the creation of better and more successful treatments, so it is possible our study includes patients who were treated with therapies that are no longer used in the standard regimen and had lower response rates than current options"), I believe that a more detailed explanation is needed to make sure that practicioners understand what the data does and does not answer so that they can make properly informed decisions for their patients. I would encourage the authors to include in the discussion the dates of FDA approval for single-agent ipilimumab; single agent nivolumab; single-agent pembrolizumab; ipilimumab + nivolumab; and nivolumab + relatlimab- and particularly to highlight the years of diagnosis and treatment of the patients in this cohort relative to those dates.
I would suggest that the data in this manuscript may provide comparative outcomes versus Ipilimumab, but without additional information about the treatments given I don't think that conclusions can be make about outcomes versus the current NCCN-recommended first-line treatment options for patients with metastatic melanoma (essentially all of the options above EXCEPT Ipilimumab) which is needed to "help" physicians who would read this paper in their decision-making.
I do think that this data is publishable so that subsequent analyses can be performed to assess comparisons versus current standards of care.
2. I would also note that a limitation of the analyses is that the only factor that is significantly associated with Overall Survival in stage IV melanoma patients, including patients with lung metastases (with or without lymph node metastases) is the serum LDH level. Serum LDH is the only validated prognostic factor in stage IV patients other than the organ sites involved, and it is included in the AJCC staging system for melanoma. The lack of this factor in the analyses should be acknowledged as another key limitation of the data, particularly for the analysis adjusted for risk factors- as none of the factors used in this analysis are used in the AJCC staging system.
Author Response
Comments 1:
The key problem with the manuscript remains that conclusions are being drawn without knowing what immunotherapies the patients received- and particularly the likelihood that the immunotherapies received by the patients in this cohort are irrelevant to contemporary patients. As a result, the conclusions made by the authors may be inaccurate in determining what is the relative efficacy of melanoma lung metastectomy "in the era of immunotherapy." Importantly, making conclusions on data that isn't "good" may lead current practicioners to make decisions about their patients that aren't truly evidence based and potentially harmful to patients.
As I explained in my initial review, based on the dates of diagnosis of the cohort, I believe that most patients treated with immunotherapy likely received single-agent ipilimumab, which has a response rate in stage IV patients of 10-15%- which is basically in the same range as the therapies that the authors noted to be part of the previous era of melanoma treatments in the paragraph that starts on line 217. While the authors do have a single sentence at the end of the Discussion acknowledging this to a degree ("Immunotherapies have changed greatly since 2010 with the creation of better and more successful treatments, so it is possible our study includes patients who were treated with therapies that are no longer used in the standard regimen and had lower response rates than current options"), I believe that a more detailed explanation is needed to make sure that practicioners understand what the data does and does not answer so that they can make properly informed decisions for their patients. I would encourage the authors to include in the discussion the dates of FDA approval for single-agent ipilimumab; single agent nivolumab; single-agent pembrolizumab; ipilimumab + nivolumab; and nivolumab + relatlimab- and particularly to highlight the years of diagnosis and treatment of the patients in this cohort relative to those dates.
I would suggest that the data in this manuscript may provide comparative outcomes versus Ipilimumab, but without additional information about the treatments given I don't think that conclusions can be make about outcomes versus the current NCCN-recommended first-line treatment options for patients with metastatic melanoma (essentially all of the options above EXCEPT Ipilimumab) which is needed to "help" physicians who would read this paper in their decision-making.
I do think that this data is publishable so that subsequent analyses can be performed to assess comparisons versus current standards of care.
Comments 2:
I would also note that a limitation of the analyses is that the only factor that is significantly associated with Overall Survival in stage IV melanoma patients, including patients with lung metastases (with or without lymph node metastases) is the serum LDH level. Serum LDH is the only validated prognostic factor in stage IV patients other than the organ sites involved, and it is included in the AJCC staging system for melanoma. The lack of this factor in the analyses should be acknowledged as another key limitation of the data, particularly for the analysis adjusted for risk factors- as none of the factors used in this analysis are used in the AJCC staging system.
Response:
Thank you very much for taking the time to review this manuscript. Please find the detailed responses below and the corresponding in track changes in the re-submitted files.
We added the limitation that serum LDH could not be assessed with a reference to the AJCC guidelines in lines 733-735. We also added the FDA approval dates with its reference and explanation on the effect of our dataset in lines 749-754. Finally, we added lines in the conclusion to emphasize our results may not be able to be interpreted for current immunotherapy treatment options and for readers to draw caution to making this conclusion. The lines are “Additionally, without being able to analyze the type of immunotherapy used for each patient, conclusions are difficult to draw between our results and the current immunotherapy regimen. However, our data may assist with the comparison between previous and current treatment options for metastatic melanoma while assessing the benefits of pulmonary metastectomy” and “and we draw caution to expanding this conclusion to current immunotherapy options with pulmonary metastectomy.”

Reviewer 2 Report (New Reviewer)
Comments and Suggestions for Authors
The authors addressed proposed comments adequately. No more comments!
Author Response
The authors addressed proposed comments adequately. No more comments!
Response:
Thank you very much for taking the time to review this manuscript.
Reviewer 3 Report (New Reviewer)
Comments and Suggestions for Authors
My previous points have been reformulated by the authors.
Author Response
My previous points have been reformulated by the authors.
Response:
Thank you very much for taking the time to review this manuscript.
This manuscript is a resubmission of an earlier submission. The following is a list of the peer review reports and author responses from that submission.
Round 1
Reviewer 1 Report
Comments and Suggestions for Authors
This is a retrospective study comparing different treatment approaches in melanoma patients with lung metastases (immunotherapy alone vs metastasectomy alone vs immunotherapy plus metastasectomy). The authors report that metastasectomy resulted in prolonged overall survival as compared to immunotherapy alone and that after adjusting for risk factors immunotherapy plus surgery was not superior to surgery alone.
Overall, the article is well written and clearly presented, however I have concerns about potential bias. It is not clear whether the patients included in the study all had a single lung metastasis or multiple metastases and in general whether the lung disease burden was balanced between different treatment groups. It is possible that patients who did not receive surgery had greater disease burden.
The type of immunotherapy received is not clearly reported (did they all receive anti-PD1?); Furthermore, it is not clear whether patients treated only with surgery received immunotherapy perhaps later, following new progression.
It would also be important to have further information on the type of surgery performed, the state of the margins in patients undergoing metastasectomy and the incidence of post-surgical complications.
Given these biases and given the demonstrated benefit of immunotherapy for the treatment of metastatic melanoma in randomized trials, I believe that authors should be extremely cautious in their conclusions.
Author Response
Thank you very much for taking the time to review this manuscript. Please find the detailed responses below and the corresponding in track changes in the re-submitted files.
One of the limitations of our study is the inability to fully account for the disease burden within our statistical analysis. While we identified patients with strictly lung-only metastasis, we did not have access to comprehensive data on the overall disease burden, such as the number, size, or location of additional metastatic sites. This limitation restricts our ability to fully evaluate how disease burden may have impacted patient outcomes. We acknowledge this as a limitation and recognize the need for future studies with more detailed data to better understand the influence of disease burden in this patient population. Another limitation of our study is the lack of access to detailed data regarding immunotherapy treatments. As a result, we were unable to confirm whether the immunotherapy regimens were consistent across all patients. This variability in treatment could potentially influence outcomes, and we recognize this as a limitation. Another limitation of our study is the absence of detailed information on the type of surgery performed, the status of surgical margins, and post-surgical complications. This lack of data prevents us from analyzing the impact of these variables on patient outcomes. We need to acknowledge the importance of such information and included this limitation in the paper.

Reviewer 2 Report
Comments and Suggestions for Authors
Interesting work on the role of surgical Treatment of lung metastasis in melanoma. The main concerns regard the statistical methods. Specifically, the lack of information regarding adjuvant immunotherapy. When treatments are compared (i.e., surgery), groups are often quite different because of a lack of randomization. Subjects with specific characteristics are more likely to have received a certain treatment than other subjects (“indication bias”). If these characteristics also affect the outcome, a direct comparison of treatments is biased, and may merely reflect the lack of initial comparability, leading to confounding. Instead of treatment, many other factors can be investigated for their causal effects. Nevertheless, as randomization is not possible, dealing with confounding is an essential step in such analyses. As common approach, it is suggested that one could select those baseline covariates about which one would be concerned if baseline imbalance existed. Likewise, estimating each subject’s probability of having their own treatment history and use these to derive IPTW can be useful to estimate the treatment–outcome association in a regression model that is weighted using the IPTWs (The Stata Journal (2004) 4, Number 4, pp. 402–420).
Author Response
Thank you very much for taking the time to review this manuscript. Please find the detailed responses below and the corresponding in track changes in the re-submitted files.
While the use of IPTW in a regression model is helpful to reduce bias, we used a multivariate analysis to mitigate the risk of bias within our data set. We acknowledge the limitation of our studying being retrospective in its design and have added this limitation to our paper.

Reviewer 3 Report
Comments and Suggestions for Authors
The manuscript entitled “The Role of Metastasectomy and Immunotherapy in the Management of Melanoma Lung Metastases: An Analysis of the National Cancer Database” addresses the important issue of metastasectomy or immunotherapy or their combination as therapy choice for melanoma lung metastases.
The study reveals a better performance of metastasectomy plus immunotherapy over either therapy alone when unadjusted data are analyzed and an advantage of metastasectomy over immunotherapy alone or in combination when adjusted data are analyzed. There are strong differences in the groups since the treatment decision is taken considering the specific situations of the patients in addition to a clear time bias since immunotherapy has increasingly been used in the time interval reported. These differences severely affect the interpretation of the data and I do not feel sufficiently competent to evaluate the statistical procedures applied. I therefore recommend to send the paper for statistical review.
Minor comments:
The authors state: “Since only patients with lung-only melanoma metastasis were included in our final synthesis, we believed that it is reasonable to assume that the code that refers to distant tumor resection reflects patients that received a pulmonary metastasectomy.” This should be verified at least by random sampling.
Author Response
Thank you very much for taking the time to review this manuscript. Please find the detailed responses below.
The assumption that only lung metastases were accounted for stems from the way data is reported in the National Cancer Database (NCDB) by individual centers. Each center reports metastatic sites according to their documentation protocols, and in our study, we relied on this reporting to ensure we selected patients with strictly lung-only metastases. However, we did not conduct a random sampling from the data pool to further verify this assumption, as the data integrity rests on the reporting practices of the contributing centers. We are amenable to having the paper sent for statistical review.

Round 2
Reviewer 1 Report
Comments and Suggestions for Authors
I my initial review I indicated that although well written and clearly presented this article has multiple biases. These include the lack of information regarding disease burden, consistency of immunotherapy received and type of surgery across different groups of patients. The authors correctly reported these factors as study limitations. However, I believe that the potential impact of these elements on the study is too high to allow interpretation of the results.
Reviewer 2 Report
Comments and Suggestions for Authors
Dear authors and Editor
I confirm my previous revision regarding the lack of adjuvant therapy assessment in the study. This represents a systematic bias in the study design.